# Atypical Sites of Lymphadenopathy after Anti-COVID-19 Vaccine: Ultrasound Features

**DOI:** 10.3390/medicina58020197

**Published:** 2022-01-27

**Authors:** Giulio Cocco, Andrea Delli Pizzi, Alessio Lino Taraschi, Andrea Boccatonda, Antonio Corvino, Claudio Ucciferri, Katia Falasca, Massimo Caulo, Jacopo Vecchiet

**Affiliations:** 1Unit of Ultrasound in Internal Medicine, Department of Medicine and Science of Aging, “G. d’Annunzio” University, 66100 Chieti, Italy; 2Department of Innovative Technologies in Medicine and Dentistry, “G. d’Annunzio” University, 66100 Chieti, Italy; andreadellipizzi@gmail.com (A.D.P.); alessiolinotaraschi@gmail.com (A.L.T.); massimo.caulo@unich.it (M.C.); 3Department of Internal Medicine, Bentivoglio Hospital, AUSL Bologna, 40010 Bentivoglio, Italy; andrea.boccatonda@gmail.com; 4Motor Science and Wellness Department, University of Naples “Parthenope”, 80133 Naples, Italy; an.cor@hotmail.it; 5Clinic of Infectious Diseases, Department of Medicine and Science of Aging, “G. d’Annunzio” University, 66100 Chieti, Italy; claudio.ucciferri@asl2abruzzo.it (C.U.); katia.falasca@unich.it (K.F.); jacopo.vecchiet@unich.it (J.V.)

**Keywords:** lymphadenopathy, atypical sites, anti-COVID-19 vaccine, ultrasound

## Abstract

*Background and Objectives*: Several authors have reported cervical and axillary lymphadenopathies as known side effects following anti-COVID-19 vaccine administration. Few data are available about atypical locations of post-anti-COVID-19 vaccine lymphadenopathy. In this investigation, we evaluated the incidence and prevalence of postvaccine lymphadenopathy ultrasound (US) features in atypical sites. *Materials and Methods*: In this retrospective study, we retrospectively selected 64 patients on whom US was performed between January and October 2021 due to COVID-19 vaccine-related lymphadenopathy. We investigated lymph node anatomical sites, presence, number, size, shape, cortical profile, hilum outline, superb microvascular imaging (SMI), and elastosonography. *Results*: A total of 170 nodes were assessed. Atypical location was demonstrated in 5/64 patients (7.8%). In all these cases, atypical nodal involvement was associated with lymphadenopathy in a typical site (axillary, supraclavicular) ipsilateral to the vaccine injection site. Two patients presented lymphadenopathy in the infraclavicular station (3.1%), one in the pectoralis major muscle (1.6%), one in the left arm (1.6%), and one in the nuchal site (1.6%). All lymphadenopathies were oval-shaped, with a median size of 0.9 ± 0.2 cm. US features included a symmetric cortex with hilum evidence (4/6, 60%), vascular signal at SMI in both the hilar region and periphery of lymph node (5/6, 83.3%), and a US elastography pattern resembling that of adjacent tissues (5/6, 83.3%). The median age of patients with lymphadenopathies in an atypical location was 23 years. The main type of vaccine associated with lymph node appearance in atypical sites was Moderna’s mRNA-1273 (60% of patients, 4/6 lymph nodes accounting for 66.7% among atypical locations). *Conclusion*: Post-COVID-19 vaccine administration lymphadenopathies in an atypical location represent an intense immune response to antigenic stimuli and they may show alarming US traits superimposed on malignant pathologies, which may complicate the patient’s clinical and diagnostic pathway. Despite no distinctive US features between reactive post-COVID-19 vaccination and malignant lymph nodes being available, careful examination of atypical lymph node locations associated with accurate knowledge of patients’ clinical background and delay of US exam to four to six weeks after vaccine injection should be considered.

## 1. Introduction

SARS-CoV-2, a virus of the family of *Betacoronaviridae*, is the etiological agent of COVID-19 disease, a severe illness with the first official case described in December 2019 in Wuhan, China, and currently verified as being responsible for about 4.8 million deaths [1,2]. Symptoms and clinical manifestations are common to systemic diseases and include cough, dyspnea, fever, anosmia, mild to severe pneumonitis with diffuse alveolar damage, and acute respiratory distress syndrome, potentially leading to death [3,4]. The progressive spread of SARS-CoV-2 all over the world, the related disease severity, and the impact on everyday life and economics have stimulated the development of vaccines with a high safety profile to limit the rising infection rates [5,6]. As it stands, different types of vaccines manufactured by pharmaceutical giants are now authorized for administration, including the best known BNT162b2, ChAdOx1, and RNA-1273 vaccines. Despite the rise of new SARS-CoV-2 variants, BNT162b2 seems to induce a powerful immune response comparable to that of the original virus [7]. The progressive increase in the number of people who have undergone vaccination has revealed common side effects associated with the procedure. Adverse events include fever, myalgias, pain at the injection site, hypercoagulability states, elevated risk of HSV infection, acute myocarditis, appendicitis, and a copious list of unconventional and usual reactions [5]. Post-COVID-19 vaccine lymphadenopathy is a common adverse event occurring in about 3–16% of patients regardless of the vaccine type administered [5]. Generally, lymph node enlargement involves axillary and/or supraclavicular nodal stations ipsilateral to the intramuscular injection site [8,9,10,11,12,13,14]. Ultrasound (US) examination is a first-line imaging modality for evaluating morphologic lymph nodes due to its common accessibility and clear depiction of soft tissues and superficial structures provided by high-frequency linear probes [12,13]. Ultrasound examination has already played a central role in the pandemic for the management of patients with pulmonary involvement and for the selection of those in need of chest computed tomography (CT) [14,15,16]. Sometimes, benign post-anti-COVID 19 vaccination lymphadenopathies may show “nonreactive” features and may mimic severe pathologies with poorer prognosis [12,13]. The diagnostic overlap between the US features of lymph node involvement in benign and malignant disease is extremely relevant in the follow-up of neoplastic patients [17,18]. In this setting, research into the peculiar characteristics associated with post-anti-COVID19 vaccine lymphadenopathy may allow the avoidance of worthless follow-up, expansive second-level diagnostic imaging, and invasive diagnostic procedures for selected patients. In this study, we describe the incidence and US features of post-anti-COVID-19 vaccine lymphadenopathies in atypical sites to clarify their potential role in patients’ diagnostic work-up and follow-up strategies.

## 2. Materials and Methods

This is an observational retrospective case series for which the authorization of the ethics committee was waived. Patient data were obtained in accordance with National Privacy Regulations (https://www.privacy-regulation.eu/en/, accessed on 1 June 2021). We retrospectively included a total of 64 consecutive patients who underwent a clinically indicated post-anti-COVID-19 US exam between January 2021 and October 2021 due to lymphadenopathy. No patients with oncohematologic or autoimmune disease were included in our study. US Protocol Ultrasound exams were performed by three dedicated sonologists, using a Canon Aplio I800 (Shimoishigami, Otawara-Shi, Japan) in combination with a linear 5 to 18 MHz array matrix probe. All the sonologists recorded the data in consensus. In detail, they assessed for each patient the presence, size, location, number, morphology (round or oval), and cortex–hilum (symmetric cortex with hilum evidence, asymmetric cortex with hilum evidence, no hilum evidence) of the lymph nodes [19]. Moreover, superb microvascular imaging (SMI) and elastosonography were evaluated [20,21]. SMI was used to investigate the presence of a centrally located vascular hilum without aberrant vascular signals, peripheral vascular signals, or central and peripheral vascular signals. Lymph node elasticity was assessed by evaluating two regions of interest (ROIs): One ROI positioned in the target region (lymph node) and the second ROI in the adjacent tissue (normal muscles or subcutaneous tissue). The strain ratio and subsequent differentiation into “soft” and “hard” were then automatically computed by the US device [21].

## 3. Results

A total of 64 patients who had received at least the first dose of anti-COVID-19 vaccine were assessed (Appendix A). The overall number of lymph nodes evaluated was 170 and 5/64 patients presented atypical lymphadenopathy locations (7.8%) along with common nodal involvement (axillary, supraclavicular) ipsilateral to vaccine injection (3 women and 2 men, 60% and 40%, respectively, among atypical sites). In detail, two patients presented an overall number of three lymph nodes (respectively 2 and 1) located in the infraclavicular station, one patient in the pectoralis major muscle (1 lymph node), one patient in the left arm (1 lymph node), and one patient in the nuchal site (1 lymph node). Atypical sites of post-anti-COVID-19 vaccine lymphadenopathy were mainly infraclavicular (2/6, 33.3%) and oval-shaped with a median size of 0.9 ± 0.2 cm. US features included a symmetric cortex with hilum evidence (4/6, 60%), vascular signal at SMI in both the hilar region and periphery of the lymph node (5/6, 83.3%), and a US elastography pattern resembling that of adjacent tissues (5/6, 83.3%). A single lymph node located in the nuchal site showed a prevalent hard pattern at US elastography (1/6, 16.7%) (Figure 1, Figure 2, Figure 3, Figure 4 and Figure 5). The median age of patients with lymphadenopathies in an atypical location was 23 years. The main type of vaccine associated with lymph node appearance in atypical sites was Moderna’s mRNA-1273 (60% of patients, 4/6 lymph nodes accounting for 66.7% among atypical locations), followed by the AstraZeneca ChAdOx1 vaccine and the Pfizer/BioNTech BNT162b2 mRNA vaccine at the same level (1 patient with 1 lymph node, respectively). The mean time of atypical lymph node appearance was 3 days after vaccine injection (±0.7 days), and the median time to lymphadenopathy resolution was 33.75 ± 6.6 days.

## 4. Discussion

Post-COVID-19 vaccine lymphadenopathies are a common adverse reaction widely described in recent literature [12,13]. This event could be underestimated, in fact, most patients underwent US examination for pain and/or palpable masses, but there are also patients who underwent US examination for asymptomatic lymph nodes found incidentally with other imaging tests (mammography, CT, or MRI) [12]. Typical lymph node sites involved are axillary and supraclavicular stations ipsilateral to injection [9,12,13,22]. Sometimes, post-anti-COVID-19 lymphadenopathies may show features superimposed on malignant pathologies, representing a potential pitfall for patients belonging to high-risk categories. In this scenario, the European Society of Breast Imaging (EUSOBI) recommends labeling as benign or probably benign all lymphadenopathies that arise in the 12 first weeks after anti-COVID-19 vaccine in the presence of negative mammography [18]. In our investigation, a consistent number of subjects demonstrated lymphadenopathies in atypical locations ipsilateral to the site of vaccine injection (5/64 patients, 7.8%), in association with more common axillary or supraclavicular nodal involvement. Significant dissimilarity in atypical lymph node sites was detected among vaccines and genders. The Moderna mRNA-1273 vaccine showed the most atypical lymph node locations (66.7%) among the three vaccines administered in our country. That fact agrees with the high prevalence of lymphadenopathies described in Moderna vaccine trials where no data about atypical locations were included [23]. The low prevalence of the involvement of atypical lymph node stations after the ChAdOx1 vaccine and the onset in a middle-aged patient could be linked to AstraZeneca suspending the use of the vaccine in Italy for people under 60 years old due to a high risk of thromboembolism [24]. Atypical lymphadenopathies were more common in women than men (60% vs. 40%, respectively). Limited data on atypical lymph node location and features following anti-COVID-19 vaccinations are currently available, especially at US. D’Auria et al. [11]., in a pictorial essay, highlighted a single subclavicular lymph node in a patient who had undergone COVID-19 vaccination. The US elastography and traits are like those reported in our investigation, with stiffness comparable to that of adjacent tissue, oval morphology and no significant color Doppler or hilar abnormalities. Likewise, Hiller and colleagues [25] detected atypical infraclavicular lymphadenopathy related to COVID-19 vaccine at US. No significant vascular abnormalities or shape alterations were described in the report. Both cases show lymph node normalization or size and morphologic stability or resolution three to four weeks after first US evaluation, an interval comparable to reports in the literature [22].

Similarly, PET/CT studies have highlighted relevant FFDG uptake in a consistent number of lymph nodes after COVID-19 mass vaccination programs [17]. Atypical nodal locations (infraclavicular, pectoralis muscle) and a high frequency of hypermetabolic lymphadenopathies in younger patients were also illustrated with the same procedure. The incidence of hypermetabolic lymphadenopathies revealed in PET-CT scans is higher in the cluster of patients with elevated anti-spike titers. This result clearly suggested a strong B-cell immune response evoked by the COVID-19 vaccine, clearly depicted along lymphatic drainage of the injection site [26]. As already mentioned, we hypothesize that US detection of lymph nodes in atypical locations could be only a manifestation of a stronger immune response spreading along lymphatics ipsilateral to vaccine injection. Analogously, powerful immune response and “reactive” lymphadenopathies develop rapidly after well-known vaccines, including human papillomavirus (HPV), Influenza A (H1N1), Bacillus Calmette–Guérin (BCG), and smallpox vaccines [27,28]. Isolated data about contralateral supraclavicular lymph node enlargement after COVID-19 vaccine injection have also been reported [29]. In this case, the hypothesis of a strong immune response elicited ipsilateral to the injection site does not explain the contralateral nodal involvement. As things stand, more data are needed to settle the reason for atypical locations of nodal enlargement post-anti-COVID-19 vaccine. In all reports cited, no definite US features are capable of distinguishing post-anti-COVID-19 vaccine-induced lymphadenopathies from lymph node involvement from malignant pathologies [30]. Despite lymph nodes often located in atypical sites showing a US elastography pattern like that of adjacent tissues, that characteristic alone is not enough to establish a benign nodal pattern. Therefore, care should be adopted in US evaluation of lymphadenopathy, especially in cancer-affected patients, and accurate knowledge of the patient’s clinical background should be mandatory. In this scenario, scientific findings and the position paper of clinical societies involved in cancer management suggest delaying US or CT by at least four to six weeks after COVID-19 vaccine injection to avoid misdiagnosis [31]. Our investigation shows several limitations. The small sample size, the descriptive and observational nature of the study, should be carefully considered in data evaluation”. Furthermore, we included in the study patients under different vaccinations and in different moments after vaccination. The patients were obtained in a single center. Due to these limits, more investigations are needed to clearly assess the real prevalence and alternative US features of post-anti-COVID-19 vaccine lymphadenopathies located in atypical sites.

## 5. Conclusions

Post-anti-COVID-19 lymphadenopathies may be in atypical locations ipsilateral to vaccine injection in association with the involvement of more common axillary and supraclavicular nodal stations. The atypical location, clinical appearance, and US features of post anti-COVID19 vaccine lymphadenopathies can sometimes simulate malignancy. Careful evaluation of the location of atypical lymphadenopathies should be considered in high-risk patients with a known history of neoplasia, to avoid misdiagnosis. A four- to six-week delay of US evaluation after anti-COVID-19 vaccine injection should be considered together with accurate knowledge of the patient’s clinical background.

## Figures and Tables

**Figure 1 medicina-58-00197-f001:**
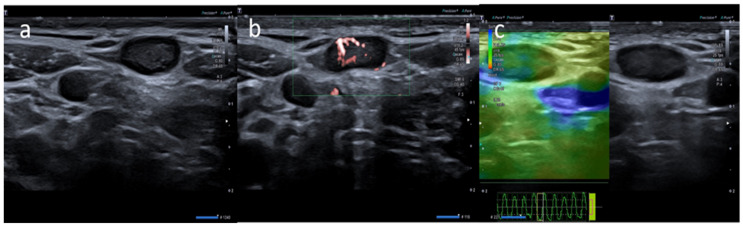
A 20-year-old male with palpable unilateral subclavicolar adenopathy noted three days after receiving the first dose of the Moderna COVID-19 vaccine in his left deltoid muscle. (**a**) B-mode sonogram image shows two ovalar hypoechoic lymph nodes with symmetric cortical thickening and hilum evidence. (**b**) SMI image shows central and peripheral vascularization. (**c**) Elastosonography strain shows a similar pattern of the node compared surrounding tissue.

**Figure 2 medicina-58-00197-f002:**
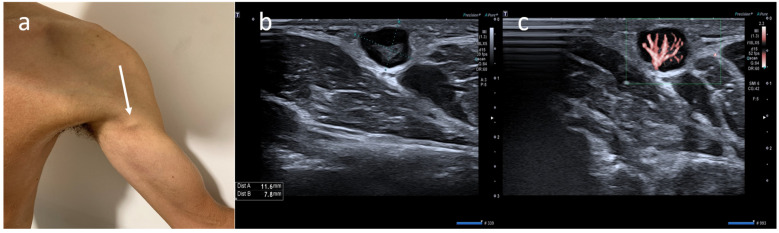
(**a**) A 21-year-old male with unilateral left arm adenopathy (white arrow) noted four days after receiving the first dose of the Moderna COVID-19 vaccine in his left deltoid muscle. (**b**) B-mode sonogram image shows ovalar lymph node with asymmetric cortex and dislocate hilum. (**c**) SMI image shows central and peripheral vascularization.

**Figure 3 medicina-58-00197-f003:**
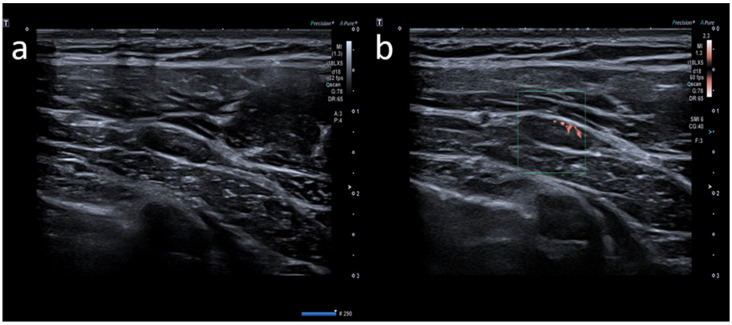
A 23-year-old female with pectoral swelling noted three days after receiving the first dose of the Pfizer COVID-19 vaccine in her left deltoid muscle. (**a**) B-mode sonogram image shows pectoral intramuscular ovalar hypoechoic lymph node with symmetric cortical thickening and hilum evidence. (**b**) SMI image shows central and peripheral vascularization.

**Figure 4 medicina-58-00197-f004:**
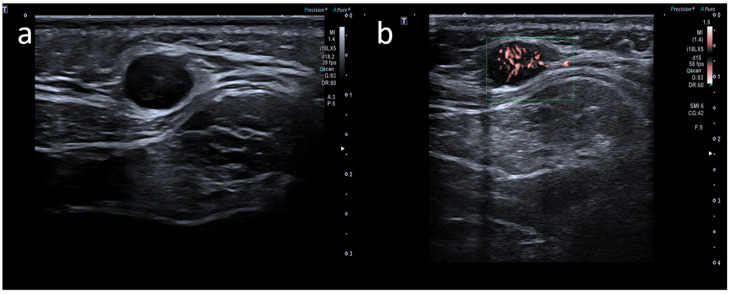
A 35-year-old male with palpable unilateral nuchal adenopathy noted three days after receiving the second dose of the Moderna COVID-19 vaccine in his left deltoid muscle. (**a**) B-mode sonogram image shows a lymph node ovalar hypoechoic with hilum absence. (**b**) SMI image shows central and peripheral vascularization.

**Figure 5 medicina-58-00197-f005:**
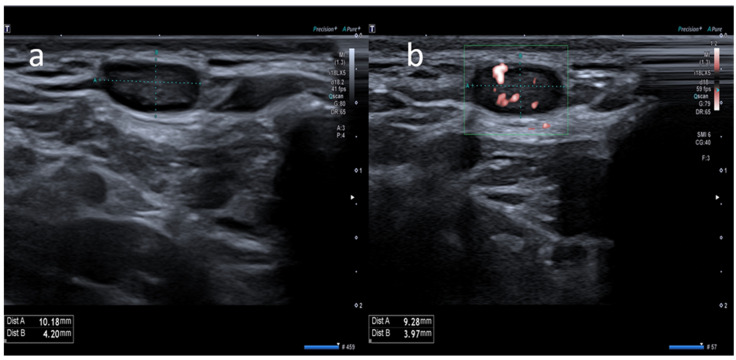
A 63-year-old female with palpable unilateral subclavicolar adenopathy noted four days after receiving the second dose of the AstraZeneca COVID-19 vaccine in her left deltoid muscle. (**a**) B-mode sonogram image shows ovalar hypoechoic lymph node with symmetric cortical thickening and hilum evidence. (**b**) SMI image shows central and peripheral vascularization.

## Data Availability

The datasets generated during and/or analyzed during the current study are not publicly available due to the clinical and confidential nature of the material but can be made available from the corresponding author on reasonable request.

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
