# Peer review of "Atypical Sites of Lymphadenopathy after Anti-COVID-19 Vaccine: Ultrasound Features"

_medicina, 2022, doi:10.3390/medicina58020197_

Round 1
Reviewer 1 Report
This article is an observational review of 5 case-studies of emerging atypical lymphadenopathies, observed by ultrasound, following administration of COVID-19 vaccines. Lymphadenopathy typically occurs in ipsilateral locations to injection site as an immune response to antigenic stimuli. They appear after a mean time of 3 days post-COVID-19 vaccine, in 3-16% of vaccinees regardless of vaccine type - although prevalent following the Moderna mRNA-1273 vaccine - and resolve in 3-4 weeks. Since their diagnostic can overlap with malignant diseases and limited data currently exists, their precise characterization could avoid thorough follow-up strategies. Because ultrasound evaluation doesn’t permit to establish a nodal benign pattern, authors emphasize on the necessity to accurately know patient’s clinical background, especially in cancer-affected patients as well as delaying ultrasound or CT up to 4-6 weeks after vaccine injection to avoid misdiagnosis.
The article is comprehensible and well written.
I suggest some minor revisions as follow:
- Line 2: The title indicate ‘ the anti-Covid19 vaccine’ however the article studies lymphadenopathies regardless of vaccine type (Moderna’s, Pfizer’s or Aztrazeneca’s). I suggest to modify by: ‘Atypical sites of lymphadenopathy after (the) anti-COVID-19 vaccines: ultrasound features’
- Figure 1 legend: line 116 indicates a ‘male’ but line 117 the pronoun is ‘her’ left muscle. Please correct.
- Figure 1b, I suggest thickening the green square or to choose a brighter color to contrast it better from the image.
- Figure 2: add b/ and c/ on the images
- Figure 2 legend: line 121 indicates a female while the pronoun used line122 is ‘his’
- Figure 2c: I suggest thickening the green square or to choose a brighter color to contrast it better from the image.
- Figure 3b: I suggest thickening the green square or to choose a brighter color to contrast it better from the image.
- Figure 4 legend: line 129 indicates a ‘male’ while the pronoun used in line 130 is ‘her’
- Figure 4b: I suggest thickening the green square or to choose a brighter color to contrast it better from the image.
- Figure 5b: I suggest thickening the green square or to choose a brighter color to contrast it better from the image. This figure actually indicates the sizes of the nodes. It would be informative to add this information on all the previous images if possible.
Author Response
Thank you so much for improving our article.

Reviewer 2 Report
Minor comments:
I would like authors to include a report/reference on the resolution of vaccine-induced lymphadenopathy to make their point that advocate delaying US or CT by at least 4–6 weeks after COVID-19 vaccine injection to avoid misdiagnosis.....
Author Response

(The authors gave the same response as above.)

Reviewer 3 Report
The authors present an interesting study investigating the atypical sites of lymphadenopathy after COVID-19 vaccination. They show a mainly descriptive study of such cases, with ultrasound (US) characteristics.
Nevertheless, the authors included a very low sample size (n=64). Although they represent patients that received ultrasound examination (that is not frequent following vaccination), given the high number of patients vaccinated and the relatively high proportion of lymphadenopathies associated with vaccination, it seems a very scarce and limited study on this topic. Moreover, the topic of interest (atypical site) was present in only 5 patients, which could be presented as a five-patient clinical case report rather than as an original article. Besides, n=64 would be reasonable if all patients received the same vaccine but, given that they received different vaccines, the sample should be considered even lower through stratification by vaccine type.
However, and despite these limitations, they show an interesting finding: all atypical sites are associated with another ipsilateral typical-site adenopathy and with young age (median 24 years), and they describe the US characteristics through images of good quality. Although I believe that the impact of this study is very limited given the high number of studies in this topic and the said limitations, I do not feel qualified to consider its rejection (the Editor should decide on this). Regarding the scientific appropriateness, I think this study complies with most of the required standards and based exclusively on this, I believe this manuscript is acceptable upon major changes that I recommend below:
INTRODUCTION
I believe that the justification and current knowledge on this topic is sufficiently approached.
MATERIALS AND METHODS
- Please delete “spontaneous” as type of study. Also, in my opinion, this seems an observational retrospective case series.
- From where were obtained the 64 patients included? From the same hospital centre? Which?
- According to some studies, ultrasound is only recommended when clinical concern persists 6 weeks after the final vaccination dose (https://10.1016/j.jacr.2021.03.001) (your reference 32). What were the causes for indicating US in your sample? It is possible that a high number of adenopathy nodes was an indicating criterium? (170 nodes of 64 patients make a mean of 2.7 nodes per patient – is that the normal manifestation of adenopathy after vaccination, involving several nodes?) I think it would be helpful to know the indications and the time from the vaccination to US test of the included patients.
RESULTS
I believe that the results are well summarised and illustrated with sufficient quality. Please just confirm that figure 2a corresponds to a female.
- Please include the exact number of patients that received each type of vaccine in your total sample. Please consider including a table summarising the sociodemographic characteristics of your sample (age, sex, comorbidities, or any fact that you may have collected) and US characteristics (means, vascular sign, etc.). If you believe that this information is not of interest, please explain. At least, age and sex of your total sample should be stated. Have you considered to compare the characteristics of patients with typical vs. atypical nodes, or patients that received different types of vaccines in bivariant analyses? Do you believe that this may be useless given the small sample size? I encourage you to do it and, depending on the results, consider including it as a supplementary file.
DISCUSSION
- I think that the seriousness of adenopathy as a side effect is not perceived equally among different populations. For example, note that adenopathy occurred in 3-16% of patients according to your introduction, based on population-based studies. However, note that, in clinicians and healthcare professionals, the number of reported adenopathy as side effect are reduced to 0.3% according to a study (https://doi.org/10.3390/vaccines10010015). Therefore, probably for healthcare workers, this symptom could be under-reported. Please consider discussing on this topic.
- If there are no specific differences between post-vaccine and malignant adenopathy, will you recommend any other performance for differential diagnosis in these cases apart from ultrasound? (e.g., breast examination https://1016/j.clinimag.2021.01.016).
- When you say that Moderna showed most atypical lymph node locations, it seems that Moderna is more associated with atypical lymph node locations. Nevertheless, this would be real if we know the percentage of patients that received each vaccine in your population (in your hospital or city). For example, if most of your population received Moderna, it is normal that most of the side effects appeared in patients that received such vaccine. Please explain this point in your Discussion.
- The limitations should be clearly enlarged “Our investigation shows several limitations. The small sample size and observational nature of the study should be carefully considered in data evaluation”. Please add the inclusion of patients under different vaccinations and in different moments after vaccination (if this is true), the descriptive (not only observational) nature of your analysis and the one-center approach.
Thank you very much for the opportunity of revising your work. I believe that you made a great effort to analyse these 64 patients and that your manuscript shows interesting US images and descriptive data. I wish you the best of luck in this manuscript and in your future research.
Author Response

(The authors gave the same response as above.)

Round 2
Reviewer 3 Report
I think the manuscript has been sufficiently improved.